# MicroRNAs of the miR-17~92 family maintain adipose tissue macrophage homeostasis by sustaining IL-10 expression

Xiang Zhang[1,2,3], Jianguo Liu[4], Li Wu[1,2,3], Xiaoyu Hu[1,2,3]*

[1]Institute for Immunology and School of Medicine, Tsinghua University, Beijing, China; [2]Tsinghua-Peking Centre for Life Sciences, Beijing, China; [3]Beijing Key Laboratory for Immunological Research on Chronic Diseases, Beijing, China; [4]Division of Infectious Diseases, Allergy and Immunology, Department of Internal Medicine, Saint Louis University School of Medicine, Saint Louis University, St. Louis, United States

**Abstract** Macrophages are critically involved in not only immune and inflammatory responses but also maintenance of metabolic fitness of organisms. Combined genetic deficiency of three clusters in the miR-17~92 family drastically shifted macrophage phenotypes toward the inflammatory spectrum characterized by heightened production of pro-inflammatory mediator TNF and diminished expression of anti-inflammatory cytokine IL-10. Consequently, macrophages residing in the adipose tissues from myeloid-specific miRNA triple knockout mice spontaneously developed inflammatory phenotypes and displayed alterations of overall physiological conditions as evidenced by obesity and compromised glucose tolerance. Mechanistically, miR-17~92 family miRNAs sustained IL-10 production by promoting transcription of the *Fos* gene, which is secondary to downregulation of *Fos* by transcription factor YY1, a direct target of miR-17~92 family miRNAs. Together, these results identified miR-17~92 family miRNAs as crucial regulators of the balance between pro- and anti-inflammatory cytokines and exemplified how macrophage-intrinsic regulatory circuit exerted impactful influence on general physiology.

*For correspondence:
xiaoyuhu@tsinghua.edu.cn

Competing interests: The authors declare that no competing interests exist.

## Introduction

MicroRNAs are a class of short non-coding RNAs that regulate gene expression post-transcriptionally in metazoan organisms. miRNAs are processed from their primary transcripts (pri-miRNAs) that are cleaved by enzymes Drosha and DiGeorge syndrome critical region gene 8 (DGCR8) into ~70 nucleotide precursors (pre-miRNAs) that are further processed by Dicer into the ~22 nucleotide long mature miRNAs (*Krol et al., 2010*). For approximately 25–40% of miRNAs, their precursors are located in close proximity to other neighboring precursors on chromatin to form miRNA clusters (*Altuvia et al., 2005*; *Kozomara and Griffiths-Jones, 2014*), which typically yield mature miRNAs with distinct seed regions that may act coordinately to achieve common functions. miR-17~92 family miRNAs consist of three paralogous miRNA clusters (*Mirc1* [called 'miR-17~92' here], *Mirc2* [called 'miR-106a~363' here], and *Mirc3* [called 'miR-106b~25' here]) that encode in total 13 distinct mature miRNAs, which can be further divided into four subfamilies (miR-17, miR-18, miR-19, and miR-25) according to their seed regions (*Figure 1—figure supplement 1A and B*; *Mendell, 2008*; *Ventura et al., 2008*). Genetic studies have shown that global deficiency of the miR-17~92 cluster results in embryonic lethality, whereas deletion of miR-106a~363 or miR-25~363 cluster yields no apparent phenotypes (*Ventura et al., 2008*). Functionally, miR-17~92 family miRNAs play an

important role in developmental processes and are regarded as oncogenes during tumorigenesis (*Mendell, 2008*; *Mogilyansky and Rigoutsos, 2013*). In the immune system, miR-17~92 family miR-NAs have been implicated in the maintenance of lymphocyte homeostasis, regulation of T follicular helper cell differentiation, and modulation of B cell development (*Kang et al., 2013*; *Lai et al., 2016*; *Xiao et al., 2008*). However, relatively little is known about the functions of miR-17~92 family miRNAs in myeloid cells including macrophages.

Macrophages play an essential role in the maintenance of tissue homeostasis and in innate immunity. In response to various environmental cues, tissue resident macrophages display a wide spectrum of phenotypes ranging from highly inflammatory ones typically observed upon infections to homeostatic ones during the processes of tissue repair. The spectrum of phenotypes could be partially attributed to the relative quantities of pro- versus anti-inflammatory mediators such as the antagonizing TNF-IL-10 pair (*Mosser and Edwards, 2008*; *Sugimoto et al., 2019*). The production of these pro- and anti-inflammatory cytokines is tightly controlled to achieve an intricate balance, and disruption of such balance has been causally linked with pathogenesis of an array of inflammatory and autoimmune diseases such as inflammatory bowel disease (IBD; *Bain and Mowat, 2014*). Moreover, the impact of these immune factors reaches far beyond the immune system evidenced by their effects on overall physiological and metabolic conditions. For example, during the development of obesity, pro-inflammatory cytokines such as TNF promote insulin resistance in adipose tissue (*Chawla et al., 2011*; *Olefsky and Glass, 2010*). In the obese adipose tissues, macrophages are the most abundant infiltrating immune cells and the major source of pro-inflammatory cytokines (*Chawla et al., 2011*; *Odegaard and Chawla, 2011*). Thus, the proper macrophage phenotypes in the adipose tissue microenvironment are critical for maintaining metabolic homeostasis.

In this study, we demonstrated that miR-17~92 family miRNAs critically controlled the balance between pro-inflammatory and anti-inflammatory phenotypes in adipose tissue macrophages (ATMs). Myeloid-specific deletion of three clusters of miR-17~92 family miRNAs twisted ATMs toward a spontaneously inflammatory phenotype characterized by diminished IL-10 and elevated TNF, leading to disrupted adipose homeostasis evidenced by obesity and compromised glucose tolerance. Mechanistically, miR-17~92 family miRNAs prevented inflammation by promoting IL-10 expression through a YY1-Fos-IL-10 regulatory circuit engaging both transcriptional and post-transcriptional mechanisms. Collectively, our results uncovered crucial roles of miR-17~92 family miRNAs in regulating macrophage inflammatory phenotypes in the context of adipose homeostasis.

## Results

### miR-17~92 family miRNAs protect mice from obesity

To investigate the function of miR-17~92 family miRNAs, we generated $Mirc2^{-/-}$ (called 'miR-106a~363$^{-/-}$' here) $Mirc3^{-/-}$ (called 'miR-106b~25$^{-/-}$' here) $Mirc1^{flox/flox}$ (called 'miR-17~92$^{flox/flox}$' here) $Lyz2$-Cre triple knockout (TKO) mice, which efficiently deleted all miRNAs in miR-17~92 family (*Figure 1—figure supplement 2A–C*). Interestingly, we found that TKO mice consistently exhibited increased body weights than age- and gender-matched $Lyz2$-Cre (WT) control mice at the age of 30 weeks (*Figure 1A* and *Figure 1—figure supplement 3A–C*). Then we isolated the visceral adipose tissue (VAT) and brown adipose tissue (BAT) from these mice and found that both the absolute weights of VAT and BAT and the percentages of VAT in TKO mice were significantly higher than those in WT mice (*Figure 1B and C* and *Figure 1—figure supplement 3D*). Furthermore, we scanned these mice with magnetic resonance imaging (MRI) and found that the total adipose tissue weight and its percentage of body weight were significantly increased in TKO mice (*Figure 1D and E*). Thus, these results indicated that miR-17~92 family miRNAs protected mice from obesity. To mimic this chronic obese phenotype, we fed TKO and WT mice with a high-fat diet (HFD). Consistent with the results obtained with animals under regular diet, body weights, VAT weights, and the VAT percentages of TKO mice were significantly higher than those of WT mice upon 20 weeks of feeding with high fat-diet (*Figure 1F–H*). Meanwhile, TKO and WT mice were scanned with MRI, and MRI quantitation indicated that both total adipose tissue weight and its percentage were significantly increased in TKO mice (*Figure 1I and J*). To further clarify the physiological role of miR-17~92 family miRNAs in ameliorating obesity, we generated mice deficient of miR-17~92 family miRNAs in the

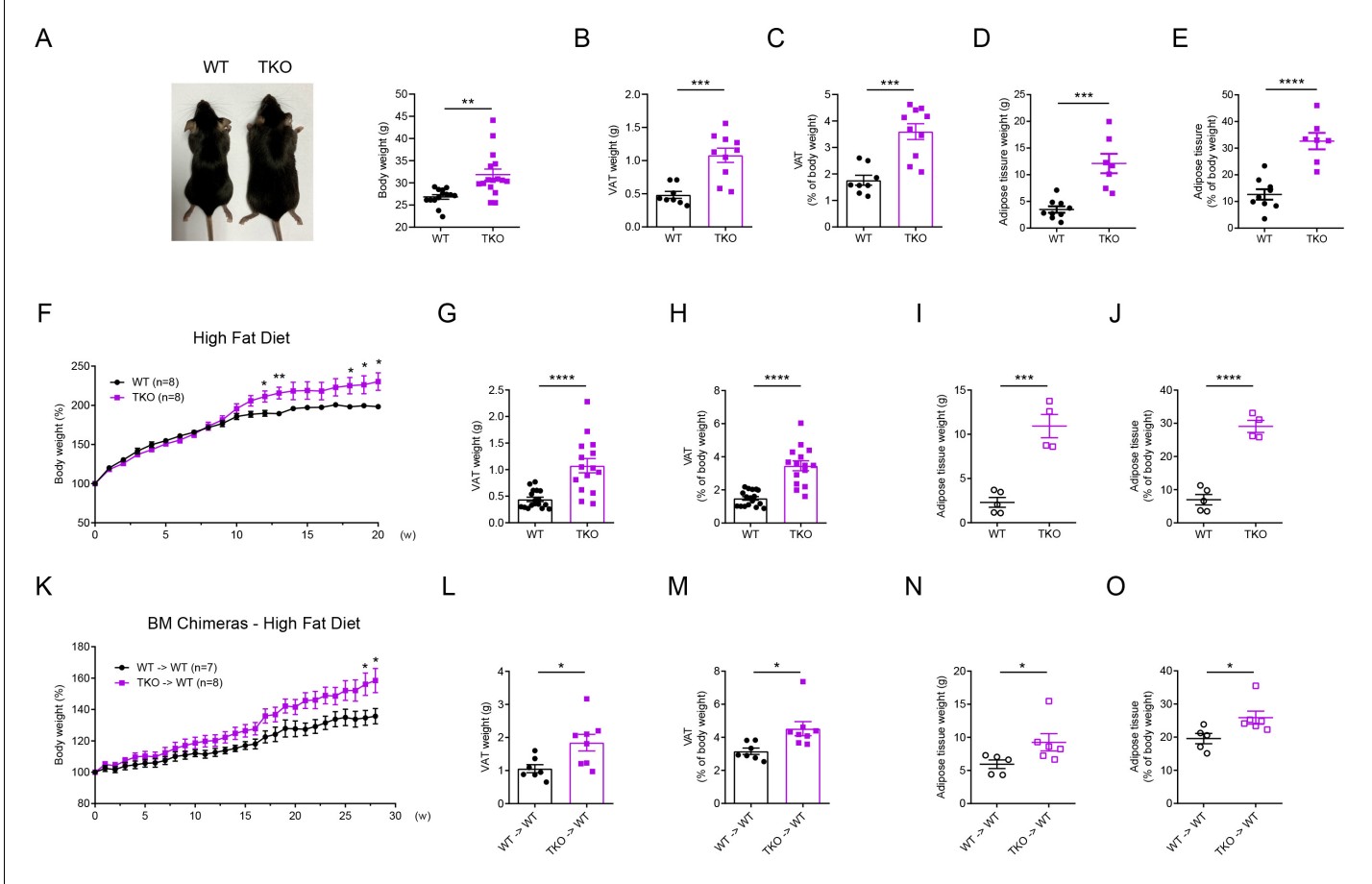

**Figure 1.** miR-17~92 family miRNAs protect mice from obesity. (**A**) A representative photograph (left panel) and body weight (right panel) of the *Lyz2*-Cre (WT) and miR-106a~363$^{-/-}$ miR-106b~25$^{-/-}$ miR-17~92$^{flox/flox}$ *Lyz2*-Cre (TKO) mice at the age of 30 weeks, n = 14–16 per group. (**B and C**) Visceral adipose tissue (VAT) weight (**B**) and VAT percentage of body weight (**C**) of WT and TKO mice at the age of 30 weeks. (**D and E**) Total adipose tissue weight (**D**) and its percentage of body weight (**E**) of WT and TKO mice were measured by scanning the whole mice with magnetic resonance imaging (MRI) machine at the age of 30 weeks. (**F–J**) WT and TKO mice were fed with a high-fat diet (HFD) and were sacrificed on 13–20 weeks post HFD. Body weight of each mouse was measured weekly, n = 8 per group (**F**); VAT weight (**G**), VAT percentage of body weight (**H**), total adipose tissue weight (**I**), and its percentage of body weight (**J**) were measured. (**K–O**) C57BL/6J mice were irradiated and transferred with WT (WT → WT) or TKO (TKO → WT) bone marrows (BMs). Then these BM chimeras were fed with an HFD and were sacrificed on 16–28 weeks post HFD. Body weight of each mouse was measured weekly, n = 7–8 per group (**K**); VAT weight (**L**), VAT percentage of body weight (**M**), total adipose tissue weight (**N**), and its percentage of body weight (**O**) were measured. *p<0.05, **p<0.01, ***p<0.001, and ****p<0.0001 (unpaired Student's *t*-test). Data are pooled from four (**A–C**), two (**D, E, F, K, L, and M**), or five (**G and H**) independent experiments (mean ± s.e.m.), or are representative of one independent experiment (**I, J, N, and O**). The online version of this article includes the following figure supplement(s) for figure 1:

**Figure supplement 1.** Schematic representation of the miR-17~92 family miRNAs.

**Figure supplement 2.** The miR-17~92 family miRNAs are efficiently deleted in TKO BMDMs.

**Figure supplement 3.** miR-17~92 family miRNAs play an important role in protection of mice from obesity.

hematopoietic compartment by transferring bone marrow (BM) cells from WT and TKO mice into irradiated C57BL/6J or CD45.1 recipients (*Figure 1—figure supplement 3E*), and subsequently fed these chimeric mice with a regular chow diet or high-fat diet. TKO chimeric mice also exhibited increased body weights compared with WT chimeric mice (*Figure 1K* and *Figure 1—figure supplement 3F*). Moreover, the absolute weights and the percentages of VAT and total adipose tissue were significantly increased in TKO chimeric mice (*Figure 1L–O*). In summary, these results demonstrate that myeloid-intrinsic miR-17~92 family miRNAs protect mice from obesity.

## miR-17~92 family miRNAs maintain adipose tissue macrophage homeostasis

We next thought to investigate the mechanisms by which miR-17~92 family miRNAs protected mice from obesity. As obesity is often associated with insulin resistance, we first examined the glucose homeostasis in TKO and WT mice with glucose tolerance test (GTT) and found that TKO mice showed higher insulin resistance than control animals (*Figure 2A*). It is well established that pro-inflammatory cytokine TNF and anti-inflammatory cytokine IL-10 are critical in regulating the insulin sensitivity and obesity (*Chawla et al., 2011*; *Olefsky and Glass, 2010*), so we first detected TNF and IL-10 levels in the serum of WT and TKO mice and found increased TNF and decreased IL-10 proteins in TKO mice relative to WT animals (*Figure 2—figure supplement 1A and B*). Then we examined the expression of *Tnf* and *Il10* mRNA in WT and TKO VAT by quantitative real-time PCR (qPCR). We found that the expression of *Tnf* was significantly higher, whereas the expression of *Il10* was significantly lower in TKO VAT than those in WT VAT (*Figure 2B* and *Figure 2—figure supplement 1C and D*) As macrophages are reported to be the major immune cells producing these cytokines in the VAT (*Chawla et al., 2011*; *Odegaard and Chawla, 2011*), we first analyzed the ATM population in VAT of WT and TKO mice. miR-17~92 family miRNAs were efficiently deleted in TKO ATMs (*Figure 2—figure supplement 2A–E*) and their deficiency did not alter the population of ATMs in mice (*Figure 2C and D* and *Figure 2—figure supplement 3A–E*). Furthermore, genome-wide RNA profiling analyses were performed with fluorescence-activated cell sorting (FACS) sorted ATMs from WT and TKO mice fed with regular chow diet or high-fat diet. Gene ontology (GO) analysis indicated that genes differentially expressed in both TKO versus WT mice and high-fat diet versus chow diet fed WT mice were enriched in the inflammatory response category (*Figure 2E* and *Figure 2—figure supplement 4A*). Meanwhile, the inflammatory score indexing cellular inflammatory phenotypes calculated based on the RNA sequencing data indicated that ATMs from TKO mice were more inflammatory than the WT counterparts under both regular chow diet and high-fat diet conditions (*Figure 2—figure supplement 4B*). Moreover, both RNA sequencing results and qPCR results showed that TKO ATMs expressed increased *Tnf* and decreased *Il10* (*Figure 2F–H* and *Figure 2—figure supplement 4C and D*). Thus, these results demonstrated that miR-17~92 family miRNA deficiency shifts ATM toward an inflammatory phenotype characterized by TNF overproduction in the absence of exogenous infectious or inflammatory signals, suggesting that miR-17~92 family miRNAs maintain ATM homeostasis to prevent obesity and to facilitate glucose tolerance.

## miR-17~92 family miRNAs balance TNF and IL-10 production in macrophages

To circumvent the technical limitation imposed by relatively scarce number of ATMs, we next utilized BM-derived macrophages (BMDMs) for further mechanistic studies. Compared to WT controls, basal expression of *Tnf* mRNA and of *Il10* mRNA was consistently upregulated and downregulated, respectively in TKO BMDMs (*Figure 3A and B* and *Figure 3—figure supplement 1A and B*), recapitulating the above observations in ATMs. Notably, due to heightened expression of these cytokine genes in TLR-activated especially lipopolysaccharide (LPS)-stimulated conditions, upregulation or downregulation of *Tnf* and *Il10* expression became more appreciable in activated macrophages than in resting cells (*Figure 3C and D* and *Figure 3—figure supplement 1C–F*). The mRNA expression patterns were further corroborated with alterations of TNF and IL-10 protein production by TKO macrophages (*Figure 3E and F*). Next, in order to clarify the connection between TNF and IL-10 in TKO BMDMs, we supplemented recombinant IL-10 protein prior to LPS stimulation and found that with sufficient quantity of IL-10, TNF production by TKO BMDMs reduced to a comparable level as WT BMDMs (*Figure 3G*). Moreover, blockade of IL-10 receptor with an anti-IL-10R antibody prior to LPS stimulation diminished the differences of TNF production observed between WT and TKO BMDMs (*Figure 3—figure supplement 1G*), implying that overexpression of TNF was likely secondary to inadequate amount of IL-10 produced by TKO cells. To functionally correct the obese phenotypes of TKO mice, an anti-TNF antibody or PBS was administrated weekly to TKO and WT mice along with HFD for 20 weeks. The anti-TNF treatment significantly reversed the obese phenotypes of TKO mice (*Figure 3H*), indicating that TNF was a functional target of miR-17~92 family miRNAs in regulating obesity. Overall, these results suggested that miR-17~92 family miRNAs suppressed TNF-mediated inflammation through promoting IL-10 expression in macrophages.

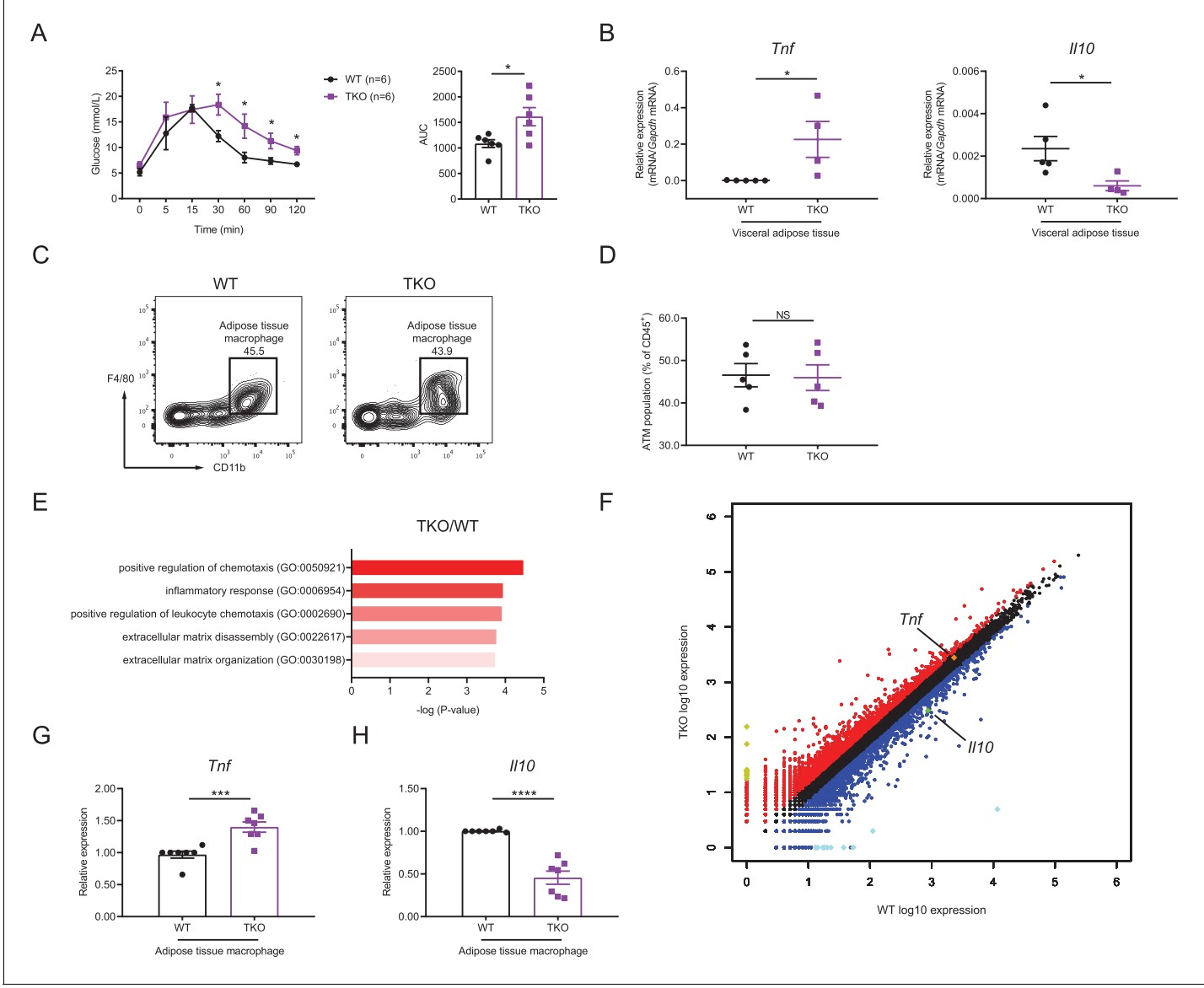

**Figure 2.** miR-17~92 family miRNAs maintain adipose tissue macrophage (ATM) homeostasis. (**A**) Glucose tolerance test in WT and TKO mice (left panel); area under curve (AUC) of left panel (right panel), n = 6 per group. (**B**) qPCR analysis of *Tnf* and *Il10* mRNA in visceral adipose tissue of WT and TKO mice. (**C and D**) Flow cytometry of ATM population in WT and TKO mice. (**C**) A representative figure showing the ATM population of CD45⁺ cells; (**D**) cumulative quantification of ATM population as in (**C**). (**E and F**) ATMs were sorted from WT and TKO mice fed with regular chow diet, RNA samples of ATMs were extracted and pooled from two mice, and genome-wide RNA profiling analyses were performed. (**E**) Gene ontology (GO) analysis of WT and TKO ATM RNA-seq datasets showing the enriched GO terms in TKO ATMs; (**F**) RNA-seq analysis showing RNA expression in TKO ATMs versus those in WT cells. RNAs upregulated in TKO ATMs were colored red, whereas RNAs downregulated were colored blue, gene *Tnf* was pointed out and colored orange, and gene *Il10* was pointed out and colored green. Top 10 upregulated genes (*Ighg2b*, *Gm11843*, *Dcdc2a*, *Mmp7*, *Slc38a5*, *Tenm2*, *Cdh16*, *Ptn*, *Ighv1-42*, and *Nanp*) were colored yellow and top 10 downregulated genes (*Csf2ra*, *Zfp125*, *Cxcl5*, *Ighv14-3*, *Igkv4-59*, *Igkv6-20*, *Igkv16-104*, *Fth-ps2*, *Igkv4-91*, and *B230303A05Rik*) were colored bright blue. (**G and H**) qPCR analysis of *Tnf* (**G**) and *Il10* (**H**) mRNA in ATMs of WT and TKO mice. Results were shown as relative expression normalized to those expressions in one WT mice of each experiment. NS, not significant (p>0.05); *p<0.05, ***p<0.001, and ****p<0.0001 (unpaired Student's *t*-test). Data are representative of three independent experiments (**C**) or are pooled from two (**A**), three (**B and D**), or five (**G and H**) independent experiments (mean ± sem).

The online version of this article includes the following figure supplement(s) for figure 2:

**Figure supplement 1.** miR-17~92 family miRNAs inhibit TNF and promote IL-10 expression in adipose tissue.

**Figure supplement 2.** The miR-17~92 family miRNAs are efficiently deleted in TKO adipose tissue macrophages (ATMs).

**Figure supplement 3.** The adipose tissue macrophage (ATM) populations are not changed in TKO mice.

**Figure supplement 4.** Adipose tissue macrophages (ATMs) in TKO mice are more inflammatory.

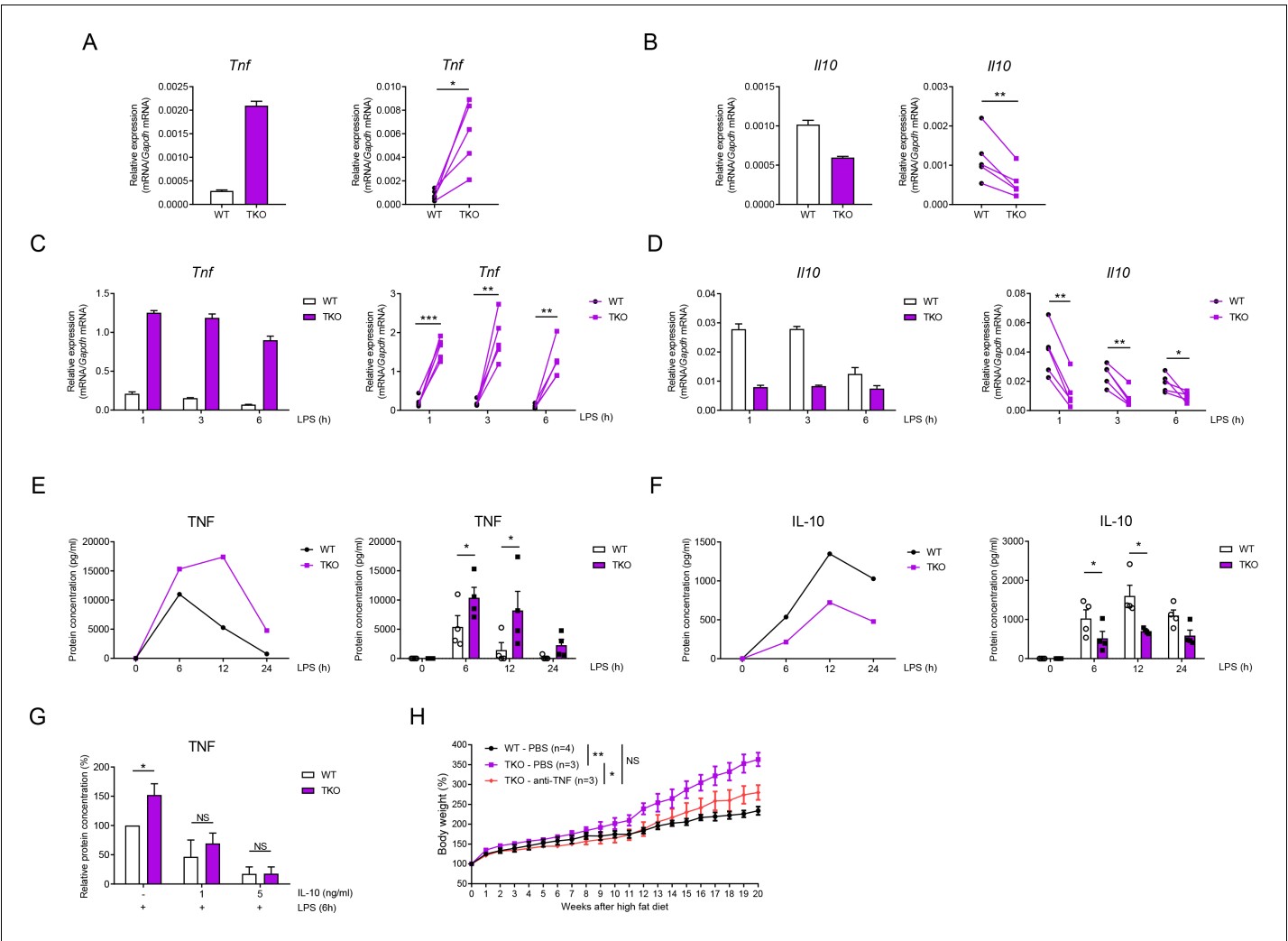

**Figure 3.** miR-17~92 family miRNAs balance the production of TNF and IL-10 in macrophages. (A and B) qPCR analysis of *Tnf* (A) and *Il10* (B) mRNA in WT and TKO BMDMs without stimulation. (C and D) qPCR analysis of *Tnf* (C) and *Il10* (D) mRNA in WT and TKO BMDMs stimulated for the indicated periods with LPS. (E and F) ELISA of TNF (E) and IL-10 (F) in supernatants from WT and TKO BMDMs stimulated for the indicated periods with LPS. (G) ELISA of TNF in supernatants from WT and TKO BMDMs pre-treated 1 hr with indicated dose of IL-10 and then stimulated with LPS for 6 hr. (H) WT and TKO mice were injected with PBS or anti-TNF (10 mg/kg) weekly and were fed with a high-fat diet. Body weight of each mice was measured weekly. NS, not significant (p>0.05); *p<0.05, **p<0.01, and ***p<0.001 (paired Student's *t*-test and two-way ANOVA). Data are representative of or are pooled from five (A–D), four (E and F), three (G), or one (H) independent experiments (mean + s.d. in A–G and mean ± sem in H).

The online version of this article includes the following figure supplement(s) for figure 3:

**Figure supplement 1.** miR-17~92 family miRNAs regulate the production of TNF and IL-10 in macrophages.

As miR-17~92 family miRNAs consist of three distinct clusters (*Figure 1—figure supplement 1A*), next we examined the relative contribution of each miRNA cluster in regulating TNF and IL-10 expression. Deletion of individual cluster or double deficiency of miR-106a~363 and miR-106b~25 clusters (DKO) did not significantly alter LPS-induced *Tnf* expression in macrophages (*Figure 4A–D*). Expression of *Il10* showed modest reduction in individual cluster single KO as well as in DKO BMDMs (*Figure 4E–H*). To investigate the collective effects of these three clusters, we compared the fold changes of *Tnf* and *Il10* expression over their respective WT controls among macrophages harboring various genotypes of miRNA deficiency and found that *Tnf* was only markedly upregulated upon deletion of all three clusters while the extent of *Il10* downregulation largely correlated with the number of deleted miRNA clusters (*Figure 4I and J*). Taken together, these results indicated that three miRNA clusters of miR-17~92 family collectively inhibited TNF and promoted IL-10 expression in macrophages.

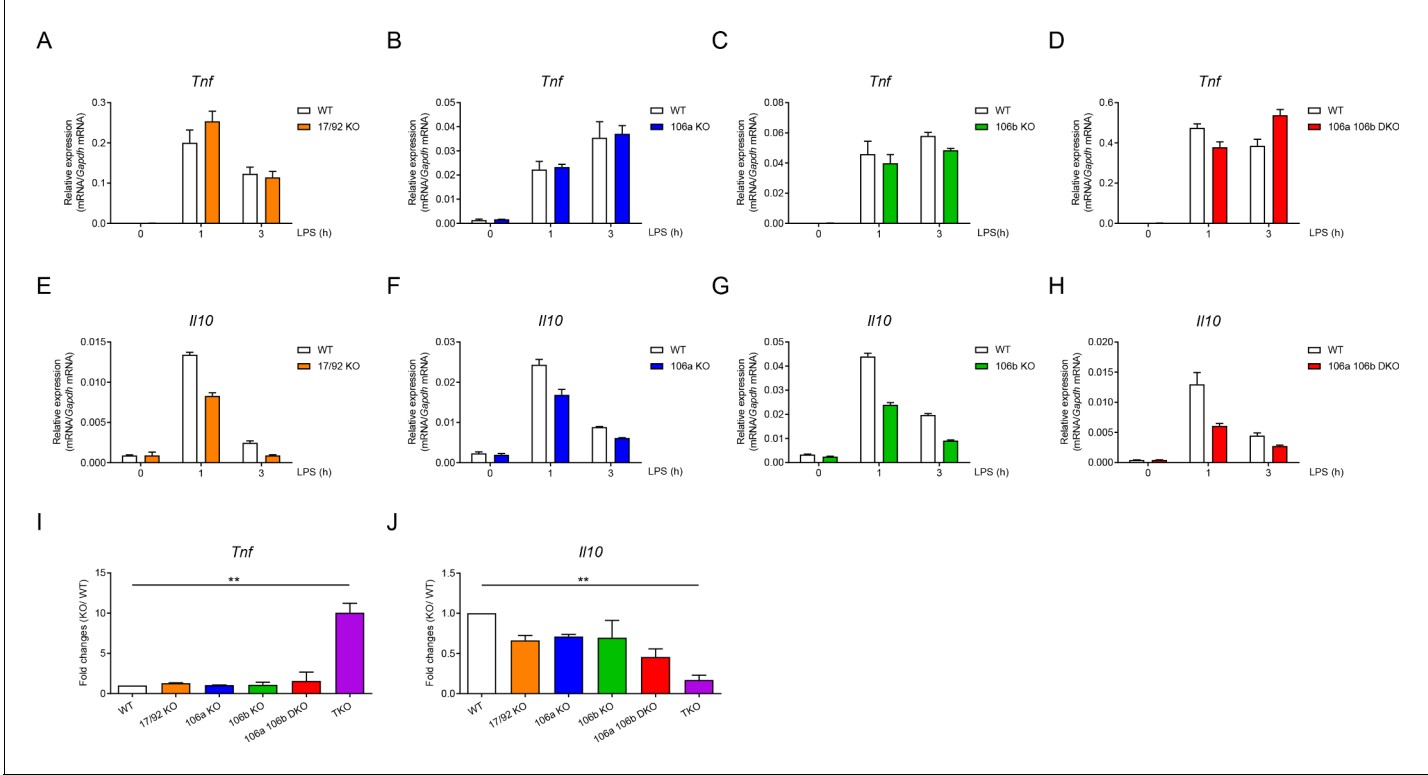

**Figure 4.** Three clusters of miR-17~92 family miRNAs regulate the expression of TNF and IL-10 collectively in macrophages. (A–D) qPCR analysis of *Tnf* mRNA in WT and miR-17~92$^{flox/flox}$ *Lyz2*-Cre (17/92 KO) (A), miR-106a~363$^{-/-}$ (106a KO) (B), miR-106b~25$^{-/-}$ (106b KO) (C), or miR-106a~363$^{-/-}$ miR-106b~25$^{-/-}$ (106a 106b DKO) (D) BMDMs stimulated for the indicated periods with LPS. (E–H) qPCR analysis of *Il10* mRNA in WT and 17/92 KO (E), 106a KO (F), 106b KO (G), or 106a 106b DKO (H) BMDMs stimulated for the indicated periods with LPS. (I and J) qPCR analysis of *Tnf* (I) and *Il10* (J) mRNA in WT and various knockout (horizontal axes) BMDMs stimulated for 1 hr with LPS; results are presented relative to those of LPS-stimulated WT BMDMs. **p<0.01 (paired Student's *t*-test). Data are representative of two (A–C and E–G) or three (D and H) independent experiments or are pooled from two to three (I and J) independent experiments (mean + s.d.).

## miR-17~92 family miRNAs promote IL-10 production via sustaining *Fos* expression

Having established IL-10 as a key target of miR-17~92 family miRNAs-mediated regulation in macrophages, we then sought to identify the mechanisms by which they promote IL-10 expression. As phosphatase and tensin homolog (PTEN) is a well-studied target of miR-17~92 family miRNAs (*O'Donnell et al., 2005*; *Xiao et al., 2008*) and PTEN and its related molecules have been implicated in modulating IL-10 expression (*Hu et al., 2006*; *Martin et al., 2005*), we first investigated whether miR-17~92 family miRNAs regulated IL-10 and TNF through targeting PTEN. Inhibition of PTEN phosphatase activity by a chemical inhibitor in TKO BMDMs did not apparently alter expression of IL-10 and TNF, suggesting that the regulation of TNF and IL-10 by miR-17~92 family miRNAs was independent of PTEN (*Figure 5—figure supplement 1A and B*). To further explore how IL-10 was regulated by miR-17~92 family miRNAs, we performed genome-wide expression profiling analysis in TKO and WT BMDMs (*Zhang et al., 2020*). Interestingly, we found that *Fos*, encoding a key component of transcriptional factor AP-1 that drives the expression of IL-10 (*Hu et al., 2006*; *Saraiva and O'Garra, 2010*), was downregulated in TKO BMDMs compared to WT controls, whereas the expression of other reported targets of miR-17~92 family miRNAs remained minimally affected (*Figure 5A*). Then we experimentally confirmed that Fos was consistently and significantly decreased in TKO BMDMs at both mRNA and protein levels (*Figure 5B–E* and *Figure 5—figure supplement 2A and B*). In contrast to Fos, the expression of Jun, another key component of AP-1 complex, did not apparently differ between WT and TKO BMDMs (*Figure 5—figure supplement 3A and B*). Furthermore, overexpression of Fos in TKO BMDMs rescued the reduced expression of

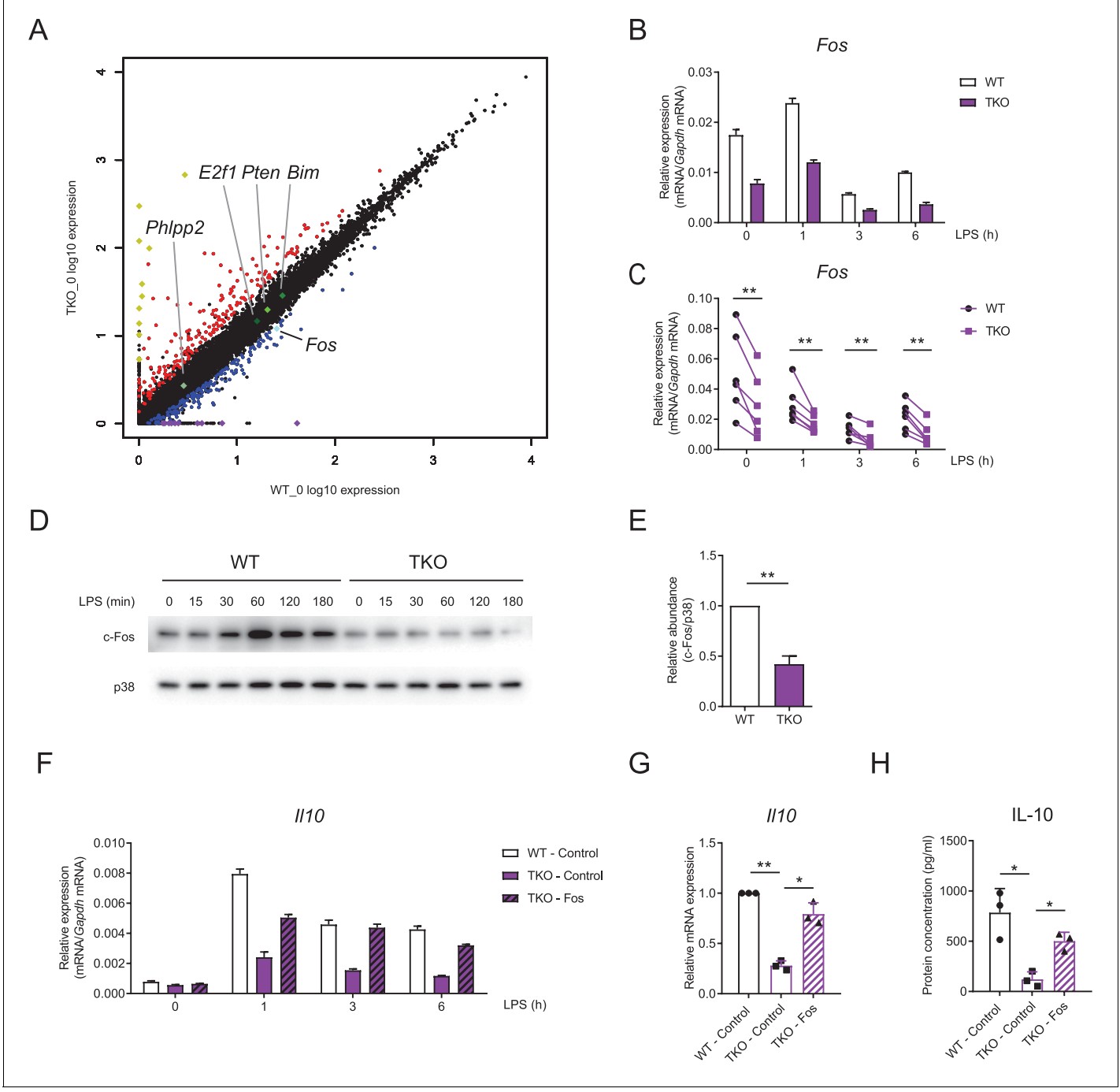

**Figure 5.** miR-17~92 family miRNAs promote the expression of Fos in macrophages. (**A**) RNA-seq analysis showing RNA expression in TKO BMDMs versus those in WT cells. RNAs upregulated in TKO BMDMs were colored red, whereas RNAs downregulated were colored blue, gene *Fos* was pointed out and colored bright blue, and genes *Phlpp2, E2f1, Pten,* and *Bim* were pointed out and colored green. Top 10 upregulated genes (*Atp6v0c-ps2, Hmga1-rs1, H2-Q6, H2-Ea-ps, Gm8580, Gm8909, Sap25, LOC547349*, H2–l, and H2–Q10) were colored yellow and top 10 downregulated genes (*Mir8114, 0610010B08Rik, Gm14430, Gm8615, Asb4, Gm14305, 6230416C02Rik, Gm38431, Pira7,* and *Cpt1b*) were colored purple. (**B and C**) qPCR analysis of *Fos* mRNA in WT and TKO BMDMs stimulated for the indicated periods with LPS. (**D**) Immunoblotting analysis of c-Fos ad p38 (loading control) in whole-cell lysates of WT and TKO BMDMs treated for the indicated periods with LPS. (**E**) Quantifications of c-Fos protein abundance in unstimulated condition in (**D**) by densitometry from four independent experiments. (**F**) qPCR analysis of *Il10* mRNA in WT and TKO BMDMs transfected with control or Fos overexpression vector and stimulated for the indicated periods with LPS. (**G**) Cumulative results from three independent experiments of *Il10* levels in LPS-stimulated 6 hr results as in (**F**), normalized to mRNA expression in control vector transfected WT cells. (**H**) ELISA of IL-10 in supernatants from WT and TKO BMDMs transfected with control or Fos overexpression vector and stimulated with LPS for 12 hr. *p<0.05 and

*Figure 5 continued on next page*

*Figure 5 continued*

\*\*p<0.01 (paired Student's *t*-test). Data are representative of six (**B**), four (**D**), or three (**F**) independent experiments or are pooled from six (**C**), four (**E**), or three (**G and H**) independent experiments (mean + s.d.).

The online version of this article includes the following figure supplement(s) for figure 5:

**Figure supplement 1.** The regulation of TNF and IL-10 by miR-17~92 family miRNAs is independent of PTEN.
**Figure supplement 2.** miR-17~92 family miRNAs sustain Fos expression in TKO BMDMs.
**Figure supplement 3.** The expression of Jun is comparable between WT and TKO BMDMs.
**Figure supplement 4.** Overexpression of Fos reduces the expression of TNF in TKO BMDMs.

IL-10 and the enhanced expression of TNF to the levels comparable with that in WT cells (*Figure 5F–H* and *Figure 5—figure supplement 4A and B*), indicating that subdued expression of IL-10 in TKO BMDMs was likely resulted from diminished Fos activities. Thus, these data demonstrated that miR-17~92 family miRNAs promoted IL-10 production via sustaining the expression of Fos.

## miR-17~92 family miRNAs directly target YY1 to regulate Fos expression

Having demonstrated that miR-17~92 family miRNAs regulate IL-10 by sustaining the expression of Fos, we next sought to investigate how Fos was regulated by miR-17~92 family miRNAs. In previous studies, Fos is reported to be inhibited by a transcriptional factor YY1 (*Shi et al., 1997*). Interestingly, despite the fact that the mRNA levels of *Yy1* were comparable between WT and TKO BMDMs, YY1 protein was significantly increased in TKO BMDMs (*Figure 6A–C*), implying post-transcriptional regulation of YY1 by miR-17~92 family miRNAs. Bioinformatics analysis with the miRanda tool (*Betel et al., 2010*) revealed YY1 as a predicted target of the miR-17 family miRNAs, a subfamily of miR-17~92 family miRNAs (*Figure 6D*). Luciferase reporter assays with the constructs containing *Yy1* 3′UTR showed significant attenuation of luciferase activities by miRNAs from miR-17~92, miR-106a~363, or miR-106b~25 cluster, whereas mutation of the miR-17 family miRNAs' binding sites on *Yy1* 3′UTR reversed such inhibitory effects (*Figure 6E*), suggesting that YY1 was likely to be the direct target of miR-17~92 family miRNAs. To further validate the involvement of YY1 in miR-17~92 family miRNAs-mediated regulation of Fos, we knocked down YY1 expression by RNA interference in WT and TKO BMDMs (*Figure 6—figure supplement 1A–C*). Knocking down YY1 significantly upregulated the expression of Fos at the mRNA and protein levels (*Figure 6F–H* and *Figure 6—figure supplement 2A and B*) and subsequently decreased TNF and increased IL-10 expression (*Figure 6I–L*). Collectively, these results suggested that miR-17~92 family miRNAs promoted Fos expression by releasing YY1-mediated inhibition.

## Discussion

In the immune system, miR-17~92 family miRNAs have been predominantly studied in adaptive immune cells including T cells and B cells. However, the functions of miR-17~92 family miRNAs in innate immune cells remain elusive. Here we illustrated the previously unidentified functions as well as underlying mechanisms of miR-17~92 family miRNAs in regulating the balance between key pro- and anti-inflammatory mediators in macrophages and uncovered a novel target of miR-17~92 family miRNAs (*Figure 6—figure supplement 3*). Such regulation bears important biological outcomes in the context of macrophage-mediated chronic inflammation with impact on general physiology such as body weight and glucose tolerance.

In recent years, due to life style changes such as increased intake of refined foods, the incidence of obesity has risen dramatically all over the globe, which in turn leads to an explosion of obesity-related disorders including but not limited to insulin resistance and type 2 diabetes, fatty liver disease, and cardiovascular diseases (*Finucane et al., 2011*; *González-Muniesa et al., 2017*). A variety of factors have been implicated in the development of obesity, including dysregulation of interconnected endocrine circuits, systemic chronic inflammation, and certain cell-intrinsic mechanisms such as oxidative stress, mitochondrial dysfunction, and endoplasmic reticulum (ER) stress (*Hotamisligil, 2006*; *Jaitin et al., 2019*; *Kahn and Flier, 2000*; *Qatanani and Lazar, 2007*). Immune modulating cytokines including TNF and IL-10 are also critical regulators of insulin signaling (*Chawla et al.,*

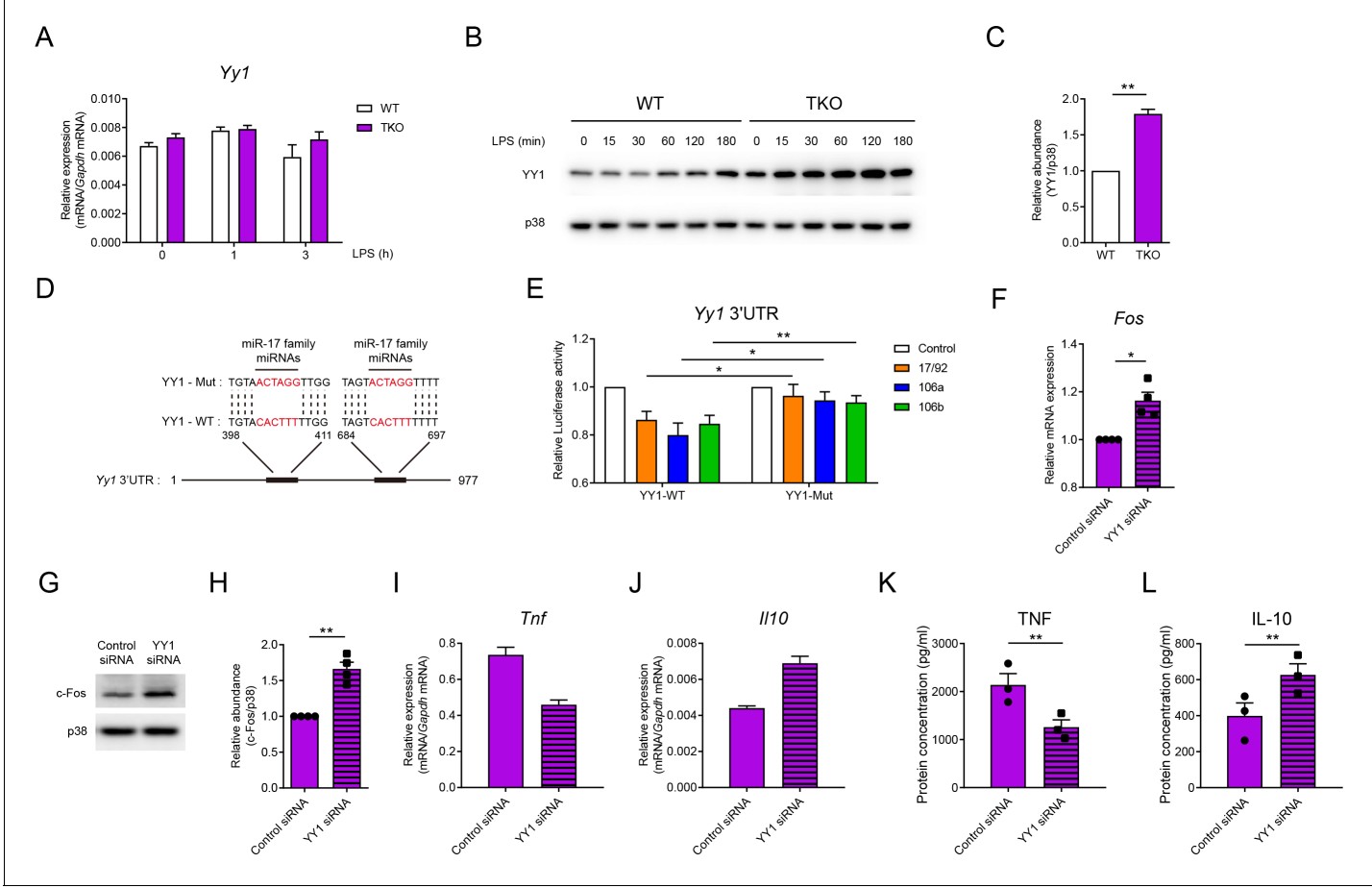

**Figure 6.** miR-17~92 family miRNAs target YY1 to promote Fos expression in macrophages. (**A**) qPCR analysis of *Yy1* mRNA in WT and TKO BMDMs stimulated for the indicated periods with LPS. (**B**) Immunoblotting analysis of YY1 and p38 (loading control) in whole-cell lysates of WT and TKO BMDMs treated for the indicated periods with LPS. (**C**) Quantifications of YY1 protein abundance in unstimulated condition in (**B**) by densitometry from three independent experiments. (**D**) Schematic illustration of predicted binding sites of miR-17 family miRNAs (red) in wild-type (YY1-WT) or mutated (YY1-Mut) *Yy1* 3′ untranslated region (UTR). (**E**) Luciferase reporter assays of *Rluc* gene expression containing the wild-type or mutated binding sites (as in **D**) in *Yy1* 3′UTR in 293 T cells co-transfected with the luciferase reporter vector and negative control (Control), miR-17~92 cluster (17/92), miR-106a~363 cluster (106a), or miR-106b~25 cluster (106b) miRNAs overexpression vector. Results are presented as Rluc/Luc activity ratio and are normalized to values in the control vector group. (**F**) qPCR analysis of *Fos* mRNA in TKO BMDMs transfected with control or YY1 short interfering RNAs (siRNAs), normalized to mRNA expression in control siRNA transfected cells. (**G**) Immunoblotting analysis of c-Fos and p38 (loading control) in whole-cell lysates of TKO BMDMs transfected with control or YY1 siRNAs. (**H**) Quantifications of c-Fos protein abundance in (**G**) by densitometry from four independent experiments. (**I and J**) qPCR analysis of *Tnf* (**I**) and *Il10* (**J**) mRNA in TKO BMDMs transfected with control or YY1 siRNAs and stimulated with LPS for 1 hr. (**K and L**) ELISA of TNF (**K**) and IL-10 (**L**) in supernatants from TKO BMDMs transfected with control or YY1 siRNAs and stimulated with LPS for 6 hr. *p<0.05 and **p<0.01 (paired Student's *t*-test). Data are representative of three (**A, B, I, and J**) or four (**G**) independent experiments or are pooled from three (**C, K, and L**), six (**E**), or four (**F and H**) independent experiments (mean + s.e.m.).

The online version of this article includes the following figure supplement(s) for figure 6:

**Figure supplement 1.** YY1 expression is efficiently knocked down by RNA interference.

**Figure supplement 2.** Fos is upregulated after knocking down YY1 in WT BMDMs.

**Figure supplement 3.** A model depicting functions and mechanisms of action of miR-17~92 family miRNAs in adipose tissue macrophages.

*2011*; *Olefsky and Glass, 2010*). For example, TNF-activated signaling molecules such as IKK, JNK, S6 kinase, and mammalian target of rapamycin (mTOR) could potentially phosphorylate insulin receptor substrate 1 (IRS1) to attenuate insulin signaling and subsequently contribute to the development of insulin resistance (*Gao et al., 2002*; *Gao et al., 2003*; *Hirosumi et al., 2002*; *Lee et al., 2007*; *Ozes et al., 2001*; *Yin et al., 1998*). Although miR-17~92 family miRNAs have been reported to regulate adipocyte development (*Chen et al., 2014*; *Wang et al., 2008*), the

current study depicted a miR-17~92 family miRNA-centric immune circuit operative in macrophages that indirectly controls metabolic phenotypes such as insulin resistance and obesity.

Previous studies have implicated the roles of miR-17~92 family miRNAs in developmental processes and tumorigenesis (*Mendell, 2008*; *Mogilyansky and Rigoutsos, 2013*), and validated multiple targets including E2F1 (*O'Donnell et al., 2005*), PTEN (*O'Donnell et al., 2005*; *Xiao et al., 2008*), Bim (*Ventura et al., 2008*; *Xiao et al., 2008*), PHLPP2 (*Kang et al., 2013*), and C/EBPβ (*Kang et al., 2020*). In this study we identified YY1 as a novel target of miR-17~92 family miRNAs. Attenuation of YY1 protein by macrophage-intrinsic miR-17~92 family miRNAs sustains expression of Fos and thus activities of a key transcription complex AP-1, resulting in optimal expression of a major homeostatic cytokine IL-10 and restraining cells from overactivation. To our knowledge, these results represent the first experimental evidence for suggesting an immune-regulatory role of YY1 in myeloid cells. The detailed analyses of regulation and function of transcription factor YY1 in macrophages warrant future investigation. Nevertheless, despite identifying YY1 as a target of miR-17~92 family miRNAs, we cannot exclude the possibility that other miRNA targets may be involved in regulating macrophage phenotypes. Further studies are needed to fully elucidate the mechanisms by which miR-17~92 family miRNAs regulate macrophage biology.

TNF is a pro-inflammatory cytokine critical for host defense against bacterial, viral, and parasitic infections, yet excessive production of TNF has been causally linked with multiple autoimmune and inflammatory diseases such as rheumatoid arthritis, IBD, and diabetes (*Bradley, 2008*; *Brenner et al., 2015*). Thus, the production of TNF must be tightly controlled by a variety of negative regulators. IL-10 is a potent anti-inflammatory cytokine that plays a central role in resolution of inflammation and robustly suppresses production of inflammatory mediators such as TNF via transcriptional and post-transcriptional mechanisms (*Moore et al., 2001*; *Ouyang and O'Garra, 2019*). Full-fledged expression of IL-10 by macrophages is achieved by coordinated action of transcription factors including c-Maf and AP-1 (*Cao et al., 2005*; *Ouyang et al., 2011*; *Saraiva and O'Garra, 2010*). Here we identified miR-17~92 family miRNAs as key positive regulators supporting IL-10 production in ATMs, raising possibilities for this family of miRNAs as potential therapeutic targets of inflammatory and metabolic disorders.

# Materials and methods

## Key resources table

| Reagent type (species) or resource | Designation | Source or reference | Identifiers | Additional information |
|---|---|---|---|---|
| Genetic reagent (*Mus. musculus*) | *Lyz2*-Cre | The Jackson Laboratory | Stock No: 004781 | |
| Genetic reagent (*Mus. musculus*) | C57BL/6J | The Jackson Laboratory | Stock No: 000664 | |
| Genetic reagent (*Mus. musculus*) | miR-106a~363$^{-/-}$ (*Mirc2$^{-/-}$*) | The Jackson Laboratory | Stock No: 008461 | |
| Genetic reagent (*Mus. musculus*) | miR-106b~25$^{-/-}$ (*Mirc3$^{-/-}$*) | The Jackson Laboratory | Stock No: 008460 | |
| Genetic reagent (*Mus. musculus*) | miR-17~92$^{flox/flox}$ (*Mirc1$^{flox/flox}$*) | The Jackson Laboratory | Stock No: 008458 | |
| Genetic reagent (*Mus. musculus*) | miR-106a~363$^{-/-}$ miR-106b~25$^{-/-}$ (*Mirc2$^{-/-}$ Mirc3$^{-/-}$*) | This paper | N/A | |
| Genetic reagent (*Mus. musculus*) | miR-106a~363$^{-/-}$ miR-106b~25$^{-/-}$ miR-17~92$^{flox/flox}$ *Lyz2*-Cre (*Mirc2$^{-/-}$ Mirc3$^{-/-}$ Mirc1$^{flox/flox}$ Lyz2*-Cre) | This paper | N/A | |
| Genetic reagent (*Mus. musculus*) | CD45.1 | Gift from Dr. Yan Shi, Tsinghua University | N/A | |
| Genetic reagent (*Mus. musculus*) | *Lyz2*-RFP | Gift from Dr. Yuncai Liu, Tsinghua University | N/A | |
| Cell line (*Homo sapiens*) | 293T | ATCC | ACS-4500 | |

*Continued on next page*

*Continued*

| Reagent type (species) or resource | Designation | Source or reference | Identifiers | Additional information |
|---|---|---|---|---|
| Antibody | Anti-mouse F4/80-APC (rat monoclonal) | eBioscience | Cat# 17-4801-82, RRID:AB_2784648 | 1:400 |
| Antibody | Anti-mouse CD45-APC/Cy7 (rat monoclonal) | BioLegend | Cat# 103116, RRID:AB_312981 | 1:400 |
| Antibody | Anti-mouse CD11b-PerCP-Cy5.5 (rat monoclonal) | eBioscience | Cat# 45-0112-82, RRID:AB_953558 | 1:400 |
| Antibody | Anti-mouse Mer-BV605 (rat monoclonal) | BD Biosciences | Cat# 747894 | 1:400 |
| Antibody | Anti-mouse CD64-PE (mouse monoclonal) | BioLegend | Cat# 139304, RRID:AB_10612740 | 1:200 |
| Antibody | Anti-mouse CD11c-FITC (hamster monoclonal) | BD Biosciences | Cat# 553801, RRID:AB_395060 | 1:400 |
| Antibody | Anti-mouse $CX_3CR1$-PE/Cy7 (mouse monoclonal) | BioLegend | Cat# 149015, RRID:AB_2565699 | 1:400 |
| Antibody | Anti-mouse Ly6C-Alexa Fluor 700 (rat monoclonal) | BioLegend | Cat# 128024, RRID:AB_10643270 | 1:400 |
| Antibody | Anti-mouse CD45.1-BV421 (mouse monoclonal) | BioLegend | Cat# 110732, RRID:AB_2562563 | 1:400 |
| Antibody | Anti-mouse CD45.2-PE/Cy7 (mouse monoclonal) | eBioscience | Cat# 25-0454-80, RRID:AB_2573349 | 1:400 |
| Antibody | p38α (C-20) (rabbit polyclonal) | Santa Cruz Biotechnology | Cat# sc-535, RRID:AB_632138 | 1:1000 |
| Antibody | c-Fos Antibody (4) (rabbit polyclonal) | Santa Cruz Biotechnology | Cat# sc-52, RRID:AB_2106783 | 1:1000 |
| Antibody | c-Jun (60A8) (rabbit monoclonal) | Cell Signaling Technology | Cat# 9165, RRID:AB_2130165 | 1:1000 |
| Antibody | YY1 antibody [EPR4652] (rabbit monoclonal) | Abcam | Cat# ab109237, RRID:AB_10890662 | 1:1000 |
| Antibody | Anti-mouse IL-10R (rat monoclonal) | Bio X Cell | Cat# BE0050, RRID:AB_1107611 | |
| Antibody | Rat IgG1 isotype control (rat monoclonal) | Bio X Cell | Cat# BE0088, RRID:AB_1107775 | |
| Antibody | Anti-mouse TNFα (rat monoclonal) | Bio X Cell | Cat# BP0058, RRID:AB_1107764 | |
| Recombinant DNA reagent | psiCHECK-2-YY1-WT (plasmid) | This paper | N/A | |
| Recombinant DNA reagent | psiCHECK-2-YY1-Mut (plasmid) | This paper | N/A | |
| Sequence-based reagent | YY1 siRNA | GenePharma | N/A | GAACUCACCUCCUGAUUAU (sense), AUAAUCAGGAGGUGAGUUC (antisense) |
| Sequence-based reagent | Primers for qPCR | This paper | N/A | see *Supplementary file 1* |
| Peptide, recombinant protein | Recombinant Murine IL-10 | PeproTech | Cat# 210–10 | |
| Commercial assay or kit | Total RNA Miniprep Purification Kit | GeneMark | TR01-150 | |
| Commercial assay or kit | Mouse TNF ELISA Set | BD Biosciences | Cat# 555268 | |
| Commercial assay or kit | IL-10 ELISA Kit | BioLegend | Cat# 431414 | |

*Continued on next page*

*Continued*

| Reagent type (species) or resource | Designation | Source or reference | Identifiers | Additional information |
|---|---|---|---|---|
| Commercial assay or kit | TaqMan microRNA Reverse Transcription Kit | Applied Biosystems | Cat# 4366596 | |
| Commercial assay or kit | TaqMan MicroRNA assays | Applied Biosystems | Cat# 4427975 | |
| Commercial assay or kit | Dual-Luciferase Reporter Assay System | Promega | Cat# E1910 | |
| Chemical compound, drug | SF1670 | Selleck | Cat# S7310 | |
| Chemical compound, drug | LPS-EB (LPS from *E. coli* O111:B4) | InvivoGen | Cat# tlrl-eblps | |
| Chemical compound, drug | Pam3CSK4 | InvivoGen | Cat# tlrl-pms | |
| Chemical compound, drug | Lipofectamine 2000 Transfection Reagent | Invitrogen | Cat# 11668027 | |
| Chemical compound, drug | *Trans*IT-TKO Transfection Reagent | Mirus | Cat# MIR2150 | |
| Software, algorithm | GraphPad Prism | GraphPad Software | RRID:SCR_002798 | |
| Software, algorithm | FlowJo(v10.0.7) | FlowJo | RRID:SCR_008520 | |
| Software, algorithm | Image J (v1.52a) | https://imagej.nih.gov/ij/index.html | RRID:SCR_003070 | |

## Mice

The miR-106a~363$^{-/-}$ (*Mirc2*$^{-/-}$, Jax stock #008461), miR-106b~25$^{-/-}$ (*Mirc3*$^{-/-}$, Jax stock #008460), and miR-17~92$^{flox/flox}$ (*Mirc1*$^{flox/flox}$, Jax stock #008458) mice were purchased from The Jackson Laboratory, which were all on C57BL/6J background. miR-17~92$^{flox/flox}$ were crossed with *Lyz2*-Cre mice to obtain mice with myeloid-specific deletion of miR-17~92 cluster. The miR-106a~363 and miR-106b~25 miRNA clusters double KO mice were obtained by crossing miR-106a~363$^{-/-}$ mice with miR-106b~25$^{-/-}$ mice. The miR-17~92, miR-106a~363, and miR-106b~25 clusters triple KO mice were obtained by crossing miR-106a~363$^{-/-}$ miR-106b~25$^{-/-}$ mice with miR-17~92$^{flox/flox}$ *Lyz2*-Cre mice. CD45.1 mice were kindly provided by Dr. Yan Shi (Tsinghua University). *Lyz2*-RFP mice were kindly provided by Dr. Yuncai Liu (Tsinghua University) which were originally bred from *Lyz2*-Cre and ROSA-tdRFP mice (*Luche et al., 2007*). Experiments were performed with mice at 6–8 weeks of age unless otherwise specified with age- and gender-matched controls. Wild-type C57BL/6J mice were used as controls for miR-106a~363$^{-/-}$, miR-106b~25$^{-/-}$, and miR-106a~363$^{-/-}$ miR-106b~25$^{-/-}$ mice, and *Lyz2*-Cre mice were used as controls for miR-17~92$^{flox/flox}$ *Lyz2*-Cre and miR-106a~363$^{-/-}$ miR-106b~25$^{-/-}$ miR-17~92$^{flox/flox}$ *Lyz2*-Cre mice. All experiments using mice were approved by the Institutional Animal Care and Use Committees at Tsinghua University (Protocol #17-HXY1).

## Cell culture and reagents

Murine BMDMs were obtained as previously described (*Xu et al., 2012*) and maintained in DMEM (HyClone) supplemented with 10% FBS (Gibco) and 10% supernatant of L929 cell as conditioned medium providing macrophage colony-stimulating factor (M-CSF, identified as complete medium). Briefly, bone marrow cells were extracted and cultured with complete medium for 5 days to derive BMDMs. Cell culture grade LPS and Pam3CSK4 were purchased from Invivogen and were used at a concentration of 10 ng/ml unless otherwise specified. SF1670 were from Selleck. anti-IL10R, anti-IgG, and anti-TNF were purchased from BioXCell. anti-IL10R or anti-IgG was added 30 min prior to LPS stimulation and were present throughout LPS exposure.

## Reverse transcription and qPCR

RNA was extracted from whole cell lysates and was reversely transcribed to cDNA as previously described (*Zhang et al., 2019*). qPCR was performed with an ABI StepOnePlus thermal cycler. Primary transcripts were measured with primers that amplify either exon–intron junctions or intronic sequences. Threshold cycle numbers were normalized to samples amplified with primers specific for glyceraldehyde-3-phosphate dehydrogenase (*Gapdh*). For qPCR analysis of mature miRNA, cDNA was prepared from total RNA, which was isolated with TRIzol reagent (Invitrogen), with the TaqMan microRNA Reverse Transcription Kit (Applied Biosystems). TaqMan MicroRNA assays were used according to the manufacturer's recommendations (Applied Biosystems) for real-time PCR. Expression of U6 snRNA was used for normalization of expression values. Primer sequences are listed in *Supplementary file 1*.

## Enzyme-linked immunosorbent assay (ELISA)

Cytokine secretion was quantified with TNF (BD Biosciences) or IL-10 (Biolegend) ELISA kit according to the manufacturers' instructions.

## BM chimeras

BM chimeras were generated as previously described (*Hu et al., 2008*). Briefly, 6-week-old recipient C57BL/6J mice or CD45.1 mice were irradiated twice at a dose of 5.5 Gy with 3 hr break in between, followed by intravenous injection of $10^6$ donor BM cells from WT or TKO mice. Chimeric mice were used for experiments 6 weeks after the initial BM transfer.

## HFD-induced obesity

Mice were fed with an HFD and body weight was measured weekly for 13–28 weeks. Mice were scanned with MRI machine (QMR06-090H, NIUMAG) for analysis of body composition before scarification for VAT excision. VAT weight was measured and was calculated for the percentage of body weight.

## Glucose tolerance test

Mice were injected intraperitoneally with glucose (2 mg/g of body weight) after fasting overnight (14–16 hr). The levels of blood glucose were measured with a glucometer (GA-3, SANNUO).

## Isolation of cells from adipose tissue

Mice were sacrificed and VAT were removed and cut into small pieces. Then tissues were digested in 5 ml ice cold digestion buffer (RPMI 1640 medium containing 10% FBS, 2 mg/ml Collagenase IV [Sigma-Aldrich], and 50 μg/ml DNaseI [Sigma-Aldrich]) for 45 min at 37°C with rotation (200 rpm). The digested tissues were passed through a 70 μm cell strainer and centrifugated at 1500 rpm for 10 min to get the isolated cells.

## Flow cytometry

Cells from adipose tissues were prepared and lysed with ACK lysing buffer (Gibco) to exclude red blood cells and were stained with antibodies on ice for 30 min in the dark. APC/Cy7 anti-mouse CD45 (30-F11, BioLegend), Percpcy5.5 anti-mouse CD11b (M1/70, eBioscience), and APC anti-mouse F4/80 (BM8, eBioscience) were used to stain ATMs, and BV605 anti-mouse Mer (108928, BD Biosciences), PE anti-mouse CD64 (X54-5/7.1 BioLegend), FITC anti-mouse CD11c (HL3, BD Biosciences), PE/Cy7 anti-mouse CX$_3$CR1 (SA011F11, BioLegend), and Alexa Fluor 700 anti-mouse Ly6C (HK1.4, BioLegend) were used to further analyze surface markers of ATMs, BV421 anti-mouse CD45.1 (A20, BioLegend) and PE/Cy7 anti-mouse CD45.2 (104, eBioscience) were used to identify ATMs from CD45.1 and CD45.2 mice, and all antibodies were used in 1:400 dilutions except CD64 (used in 1:200 dilutions). Cells were washed three times and they were performed on FACSFortessa or FACSAria III flow cytometer (BD Biosciences) and analyzed with FlowJo software (Tree Star).

## Immunoblotting analysis

Whole cell lysates were prepared as described previously (*Xu et al., 2012*). For immunoblotting analysis, lysates were separated by 10% SDS-PAGE and transferred to a PVDF membrane (Millipore)

for probing with antibodies. The antibodies against p38 (sc-535) and c-Fos (sc-52) were purchased from Santa Cruz Biotechnology, YY1 antibody (ab109237) was purchased from Abcam, and c-Jun antibody (#9165) was obtained from Cell Signaling Technology.

## RNA sequencing and analysis

Total RNA was isolated with TRIzol reagent (Invitrogen) from whole cell lysates of ATMs from WT and TKO mice fed with regular chow diet or HFD. RNA was isolated, libraried, and sequenced with Illumina Novaseq platform by Novogene (Novogene; Beijing, China). Total reads were cleaned and mapped to the mm10 reference genome and then were normalized as fragments per kilobase of transcript per million mapped reads (FPKM). Upregulated and downregulated genes in TKO ATMs were identified as p≤0.05, (FPKM +1) fold changes (TKO/WT) ≥1.6 for upregulated genes, and (FPKM +1) fold changes (TKO/WT) ≤0.6 for downregulated genes. Differential expressed genes (DEGs) between WT-HFD and WT ATMs were defined as p≤0.05 and absolute value of $\log_2$ ratio (WT-HFD/WT) ≥1. GO analysis was conducted with upregulated and downregulated genes in TKO versus WT ATMs group and DEGs in WT-HFD versus WT ATM group by Enrichr (*Chen et al., 2013*; *Kuleshov et al., 2016*). Inflammatory score was calculated with the following formula: Inflammatory score = sum of log (FPKM+1) of representative pro-inflammatory genes (*Tnf*, *Il6*, *Cxcl1*, *Ccl2*, and *Ccl8*) − sum of log (FPKM+1) of representative anti-inflammatory genes (*Il10*, *Il4*, *Il5*, and *Il13*). RNA-seq datasets with WT and TKO BMDMs from our previous published work (*Zhang et al., 2020*) were reanalyzed and plotted.

## Luciferase reporter assay

The psiCHECK2 (Promega) reporter plasmid was cloned with 3′-UTR fragments of *Yy1* containing wild-type or mutated predicted binding sites of miR-17 family miRNAs (*Supplementary file 2*) to generate *Yy1* reporter plasmids. A total of 293 T cells were plated into 24-well plates at $1 \times 10^5$ cells per well 24 hr before transfection with 10 ng reporter plasmid and 500 ng overexpression vectors encoding miRNAs of miR-17~92, miR-106a~363, or miR-106b~25 clusters using the Lipofectamine 2000 transfection reagent (Invitrogen). Luciferase assays were performed 48 hr post-transfection using the Dual-Luciferase Reporter Assay System (Promega) following the manufacturer's protocol. The renilla firefly luciferase (Rluc) activity was normalized by the firefly luciferase activity (Luc) and expression is presented as Rluc/Luc ratio.

## Cell lines

A total of 293 T cells (ATCC ACS-4500) were maintained in DMEM supplemented with 10% FBS and 1% Penicillin–Streptomycin (Gibco) and tested negative for mycoplasma.

## RNA-mediated interference

Small interfering RNA (siRNA) specifically targeting mouse *Yy1* (5′-GAACU CACCU CCUGA UUAU-3′ (sense)/5′-AUAAU CAGGA GGUGA GUUC-3′ (antisense)) and nontargeting control siRNA were from GenePharma. The siRNAs were transfected into mouse BMDMs through the use of TransIT TKO transfection reagent according to the manufacturer's instructions (Mirus Bio). Cells were then lysed for mRNA and protein extraction 48 hr post-transfection.

## Statistical analysis

p-values were calculated with a two-tailed paired or unpaired Student's t-test or two-way ANOVA. Not significant, p>0.05; *p<0.05; **p<0.01; ***p<0.001; and ****p<0.0001.

## Data and materials availability

All data supporting the findings of this study are presented within the article and its supplementary files. The RNA-seq data set is deposited in the National Center for Biotechnology Information Gene Expression Omnibus under accession number GSE129613 (*Zhang et al., 2020*) and GSE158627.

## Acknowledgements

We thank Y Zhang (University of Maryland) and B Zhang (Tsinghua University) for help with RNA-seq analysis and Y Shi (Tsinghua University) and Y Liu (Tsinghua University) for kindly providing several mouse strains. This research was supported by National Natural Science Foundation of China grants (31725010 and 31821003 to XH and 31330027 to LW), funds from Tsinghua-Peking Center for Life Sciences (XZ, XH, and LW), and funds from Institute for Immunology at Tsinghua University (XH and LW).

## Additional information

### Funding

| Funder | Grant reference number | Author |
|---|---|---|
| National Natural Science Foundation of China | 31821003 | Xiaoyu Hu |
| National Natural Science Foundation of China | 31725010 | Xiaoyu Hu |
| National Natural Science Foundation of China | 31330027 | Li Wu |
| Tsinghua-Peking Center for Life Science | | Xiang Zhang<br>Li Wu<br>Xiaoyu Hu |
| Institute for Immunology at Tsinghua University | | Li Wu<br>Xiaoyu Hu |

The funders had no role in study design, data collection and interpretation, or the decision to submit the work for publication.

### Author contributions

Xiang Zhang, Conceptualization, Data curation, Formal analysis, Validation, Methodology, Writing - original draft, Writing - review and editing; Jianguo Liu, Li Wu, Resources; Xiaoyu Hu, Conceptualization, Supervision, Funding acquisition, Investigation, Writing - original draft, Project administration, Writing - review and editing

### Author ORCIDs

Xiang Zhang ⓘD https://orcid.org/0000-0002-4729-0581
Xiaoyu Hu ⓘD https://orcid.org/0000-0002-4289-6998

### Ethics

Animal experimentation: All experiments using mice were approved by the Institutional Animal Care and Use Committees at Tsinghua University (Protocol #17-HXY1).

### Decision letter and Author response

Decision letter https://doi.org/10.7554/eLife.55676.sa1
Author response https://doi.org/10.7554/eLife.55676.sa2

## Additional files

### Supplementary files

- Supplementary file 1. Primers used in this study.
- Supplementary file 2. *Yy1* 3′UTR cloned fragments with miR-17 family miRNAs binding sites for luciferase reporter assays.
- Transparent reporting form

## Data availability

Sequencing data have been deposited in GEO under accession code GSE129613 and GSE158627.

The following datasets were generated:

| Author(s) | Year | Dataset title | Dataset URL | Database and Identifier |
|---|---|---|---|---|
| Zhang X | 2019 | RNAseq to profile transcriptomes in bone marrow-derived macrophages from TKO(mir-106a∽363-/- mir-106b∽25-/- mir-17∽92flox/flox Lyz2-Cre) and WT (Lyz2-Cre) mice | https://www.ncbi.nlm.nih.gov/geo/query/acc.cgi?acc=GSE129613 | NCBI Gene Expression Omnibus, GSE129613 |
| Zhang X | 2020 | RNAseq to profile transcriptomes in adipose tissue macrophages (ATMs) from TKO(mir-106a∽363-/- mir-106b∽25-/- mir-17∽92flox/flox Lyz2-Cre) and WT(Lyz2-Cre) mice fed with regular chow diet or high fat diet | https://www.ncbi.nlm.nih.gov/geo/query/acc.cgi?acc=GSE158627 | NCBI Gene Expression Omnibus, GSE158627 |

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
