## [Decision Letter]

**Acceptance summary:**

Your study identified novel roles of miR-17~92 family miRNAs in regulating macrophage inflammatory phenotypes. Disruption of the miR-17~92 family leads to increased pro-inflammatory activity of adipose macrophages. Myeloid specific deletion of three clusters of miR-17~92 family miRNAs pushed adipose tissue macrophages towards a spontaneously inflammatory phenotype characterized by diminished IL-10 and elevated TNF, leading to disrupted adipose homeostasis evidenced by obesity and compromised glucose tolerance. Mechanistically, miR-17~92 family miRNAs prevented inflammation by promoting IL-10 expression through a YY1-Fos-IL-10 regulatory circuit.

**Decision letter after peer review:**

Thank you for submitting your article "MicroRNAs of the miR-17~92 family maintain adipose tissue macrophage homeostasis by sustaining IL-10 expression" for consideration by *eLife*. Your article has been reviewed by Satyajit Rath as the Senior Editor, a Reviewing Editor, and two reviewers. The reviewers have opted to remain anonymous.

The reviewers have discussed the reviews with one another and the Reviewing Editor has drafted this decision to help you prepare a revised submission.

Summary:

The results presented here identified novel roles of miR-17~92 family miRNAs in regulating macrophage inflammatory phenotypes. The authors show that disruption of the miR-17~92 family leads to increased pro-inflammatory activity of adipose macrophages. Myeloid specific deletion of three clusters of miR-17~92 family miRNAs altered twisted adipose tissue macrophages towards a spontaneously inflammatory phenotype characterized by diminished IL-10 and elevated TNF, leading to disrupted adipose homeostasis evidenced by obesity and compromised glucose tolerance. Mechanistically, miR-17~92 family miRNAs prevented inflammation by promoting IL-10 expression through a YY1-Fos-IL-10 regulatory circuit. Overall, this is an interesting study (especially the fact that miR-17~92 family KO mice are "obese") with strong physiological significance and compelling molecular mechanisms. However, more experiments are needed to support a clear link between the inflammatory status observed in miR-17~92 family KO mice and adipose macrophages.

Essential revisions:

The majority of experiments was performed with BM-derived macrophages cultured for an unknown time in MCSF media. However, monocyte-derived macrophages are significantly different compared to tissue macrophages and show similarities – at the utmost – to peritoneal infiltrates after thioglycolate injection, but not to homeostatic tissue-resident macrophages (Gosselin et al., 2014). It is therefore questionable if the here presented findings on (partially LPS-stimulated) BMDMs can be translated to tissue-resident adipose macrophages under physiological conditions. It is similarly possible that the development of exacerbated weight gain in TKO is the consequence of a peripheral phenotype (for instance of monocytic origin), as indicated by the BM transplantation experiments, rather than the consequence of a defect in tissue-resident macrophages. Accordingly, the authors need to perform more experiments to show that the phenotype is caused by adipose macrophages. For example, showing the expression levels of all miR-17~92 family members in ex vivo isolated ATM subpopulations. In this regard, am important study to consider will be Jaitin et al., 2019.

It is further not clear, why the authors included LPS stimulation in their experimental design, when the phenotype is evident in TKO mice under physiological conditions. It would be easier for the reader, if the authors would focus on unstimulated cells rather than showing different time points after LPS treatment. Results from the in vitro LPS experiments cannot be extrapolated to the physiological in vivo situation. Finally, to overcome the correlative nature of the experiments, it would be necessary to show, that the "obese" phenotype can be rescued in TKO Tnf+/- mice.

The authors showed that IL-10 supplementation can rescue the TNF phenotypes in miR-17~92-deficient microphages. It will be important to further extend this result by using an IL-10 blocking system. On a similar point, in respect to Figure 6, the link between YY1-Fos-Il10 is not completely clear. The authors should show that silencing of YY1 in TKO leads to decreased Tnf and increased Il10 expression. The authors should also test the effects of YY1 loss on cytokine production?

In Figure 1D the authors show body weight during HFD and indicate higher weight of TKO. However, in Figure 1A the authors show that TKO already have higher weight under steady state conditions. The authors need to add the weights of untreated WT and TKO to the graph. Same for Figure 1I, the authors should show the weight of chimeras without HFD.

RNA-seq experiment should be performed with ex vivo isolated ATM and not with in vitro derived cells. Two time points for both genotypes are necessary ("pre-obese" and "obese") to observe gene changes during obesity development.

[Editors' note: further revisions were suggested prior to acceptance, as described below.]

Thank you for submitting your article "MicroRNAs of the miR-17~92 family maintain adipose tissue macrophage homeostasis by sustaining IL-10 expression" for consideration by *eLife*. Your article has been reviewed by Satyajit Rath as the Senior Editor, a Reviewing Editor, and two reviewers. The reviewers have opted to remain anonymous.

The reviewers have discussed the reviews with one another and the Reviewing Editor has drafted this decision to help you prepare a revised submission.

Summary:

The authors significantly improved their work during the revision and strengthen their main points with new experimental evidence significantly. However, some few minor things remain to be clarified before we can proceed to acceptance.

Essential revisions:

1) The bioinformatical analysis or the description of the corresponding RNA-Seq experiments is still complicated to understand. The authors state in their new version: "Furthermore, genome wide RNA profiling analyses were performed with fluorescence-activated cell sorting (FACS) sorted ATMs from WT and TKO mice fed with regular chow diet or high fat diet." However, in the Figure legend it is stated: "(E) Gene Ontology analysis of WT and TKO ATM RNA-seq datasets showing the enriched gene ontology terms in TKO ATMs. (F) RNA-seq analysis showing RNA expression in TKO ATMs versus those in WT cells. RNAs up-regulated in TKO ATMs were colored red, whereas RNAs down-regulated were colored blue, gene Tnf was pointed out and colored orange and gene Il10 was pointed out and colored green." Is this data shown in 2E+F now derived from obese or pre-obese animals? If this is derived from just one condition (obese or pre-obese mice), why didn't the authors show all conditions?

2) Furthermore, how many animals were used for the RNA-Seq experiment depicted in Figure 2E and F? Is this just 2 samples (one WT and one TKO) pooled from multiple mice? Please either indicate in the figure legends or in the Materials and methods section.

3) The authors stated that red/blue marked genes in Figure 2F indicates up- or down-regulated genes. What is their definition for up- or down-regulation? 2-fold change? Or is this based on statistics (which might be problematic with n=1).

4) Please highlight the top 10 differential expressed genes in the Figure 2E and Figure 5A scatter blot.

---

## [Author Response]

[…]Essential revisions:The majority of experiments was performed with BM-derived macrophages cultured for a unknown time in MCSF media. However, monocyte-derived macrophages are significantly different compared to tissue macrophages and show similarities – at the utmost – to peritoneal infiltrates after thioglycolate injection, but not to homeostatic tissue-resident macrophages (Gosselin et al., 2014). It is therefore questionable if the here presented findings on (partially LPS-stimulated) BMDMs can be translated to tissue-resident adipose macrophages under physiological conditions. It is similarly possible that the development of exacerbated weight gain in TKO is the consequence of a peripheral phenotype (for instance of monocytic origin), as indicated by the BM transplantation experiments, rather than the consequence of a defect in tissue-resident macrophages. Accordingly, the authors need to perform more experiments to show that the phenotype is caused by adipose macrophages. For example, showing the expression levels of all miR-17~92 family members in ex vivo isolated ATM subpopulations. In this regard, am important study to consider will be Jaitin et al., 2019.

We thank the reviewer for raising these valid concerns regarding utilization of BMDMs and the contribution of ATMs versus peripheral monocytes. We used BMDMs mainly for the mechanistic studies due to the scarcity of the number of ATMs (approximately 40,000 cells from 0.1 g fat tissues per mouse). To corroborate the findings obtained from in vitro cultured macrophages, we ex vivo isolated ATMs and showed that miR-17~92 family miRNAs were efficiently deleted in ATMs from TKO mice (new Figure 2—figure supplement 2C-E, described in subsection “miR-17~92 family miRNAs maintain adipose tissue macrophage homeostasis”). Importantly, in ATMs, the expression of IL-10 and TNF was regulated by miR-17~92 family miRNAs in a similar trend to that in BMDMs (new Figure 2F-H, described in subsection “miR-17~92 family miRNAs maintain adipose tissue macrophage homeostasis”), further validating BMDMs as a suitable in vitro system for the mechanistic investigations. As the *Lyz2*-Cre line used in this study does not discriminate between monocytes and certain tissue resident macrophages, we cannot definitively exclude the effects of recruited monocytes, although the robust deletion of miR-17~92 family miRNAs as well as the inflammatory phenotypes of ATMs supported the notion that ATMs might contribute significantly to the overall phenotypes. We also included the Jaitin et al., reference in the revised manuscript.

It is further not clear, why the authors included LPS stimulation in their experimental design, when the phenotype is evident in TKO mice under physiological conditions. It would be easier for the reader, if the authors would focus on unstimulated cells rather than showing different time points after LPS treatment. Results from the in vitro LPS experiments cannot be extrapolated to the physiological in vivo situation. Finally, to overcome the correlative nature of the experiments, it would be necessary to show, that the "obese" phenotype can be rescued in TKO Tnf+/- mice.

In the absence of activating signals, resting BMDMs expressed nearly undetectable amounts of inflammatory genes such as *Tnf* at both mRNA and protein levels (threshold cycle number > 30 by qPCR for *Tnf* mRNA and protein concentrations < 5 pg/ml as measured by TNF ELISA). Thus, in order to reveal the effects of miR-17~92 family miRNAs and to perform mechanistic experiments in vitro, we used LPS stimulation to activate macrophages and to upregulate inflammatory gene expression. Of note, the results obtained from activated BMDMs are in large consistent with the ex vivo isolated ATMs regarding inflammatory gene regulation.

For the phenotype rescue experiment, we agree with the reviewer that TNF loss-of-function experiment would be of great value. However, incorporating a *Tnf* null allele into the current TKO background would result in knockin and knockout of 5 different gene alleles and the time to generate such mice might take one year, to the conservative estimate. Alternatively, we employed in vivo administration of an anti-TNF antibody and found that TNF blockade markedly ameliorated the obese phenotype in TKO animals (new Figure 3H, described in subsection “miR-17~92 family miRNAs balance TNF and IL-10 production in macrophages”). We believe that this approach is equally valuable at demonstrating a role for TNF in development of obesity in TKO animals, with the added benefits of boosting therapeutic potentials.

The authors showed that IL-10 supplementation can rescue the TNF phenotypes in miR-17~92-deficient microphages. It will be important to further extend this result by using an IL-10 blocking system. On a similar point, in respect to Figure 6, the link between YY1-Fos-Il10 is not completely clear. The authors should show that silencing of YY1 in TKO leads to decreased Tnf and increased Il10 expression. The authors should also test the effects of YY1 loss on cytokine production?

The expression of IL-10 is reduced in TKO macrophages compared to that in WT macrophages, and thus we supplemented IL-10 instead of blocking IL-10 signaling. Nevertheless, we performed the loss-of-function experiment with an anti-IL-10R antibody and found that blockade of IL-10 signaling normalized the difference of TNF production in WT and TKO BMDMs (new Figure 3—figure supplement 1G, described in subsection “miR-17~92 family miRNAs balance TNF and IL-10 production in macrophages”), further supporting a role for IL-10 in the system. Moreover, silencing YY1 in TKO BMDMs led to decreased TNF and increased IL10 expression at the mRNA and protein levels (new Figure 6I-L, described in subsection “miR-17~92 family miRNAs directly target YY1 to regulate Fos expression”), agreeing with the model of miR-17~92 family miRNAs-IL-10-TNF regulatory axis.

In Figure 1D the authors show body weight during HFD and indicate higher weight of TKO. However, in Figure 1A the authors show that TKO already have higher weight under steady state conditions. The authors need to add the weights of untreated WT and TKO to the graph. Same for Figure 1I, the authors should show the weight of chimeras without HFD.

We have added the weights of WT and TKO mice with regular chow diet (new Figure 1—figure supplement 3C, described in subsection “miR-17~92 family miRNAs protect mice from obesity”) as well as the weights of WT and TKO chimeras with regular chow diet (new Figure 1—figure supplement 3F, described in subsection “miR-17~92 family miRNAs protect mice from obesity”).

RNA-seq experiment should be performed with ex vivo isolated ATM and not with in vitro derived cells. Two time points for both genotypes are necessary ("pre-obese" and "obese") to observe gene changes during obesity development.

We thank the reviewer for suggesting this experiment. We have performed the RNA-seq experiment with ex vivo isolated ATMs from ‘pre-obese’ (regular chow diet) and ‘obese’ (high fat diet for 8 weeks) mice. Analyses of gene expression data sets revealed that ATMs from both ‘pre-obese’ and ‘obese’ TKO mice showed more prominent inflammatory gene signature than those from WT mice (new Figure 2E and F and new Figure 2—figure supplement 4A and B, described in subsection “miR-17~92 family miRNAs maintain adipose tissue macrophage homeostasis”), further confirming the inflammatory phenotypes of TKO ATMs.

[Editors' note: further revisions were suggested prior to acceptance, as described below.]

[…]Essential revisions:1) The bioinformatical analysis or the description of the corresponding RNA-Seq experiments is still complicated to understand. The authors state in their new version: "Furthermore, genome wide RNA profiling analyses were performed with fluorescence-activated cell sorting (FACS) sorted ATMs from WT and TKO mice fed with regular chow diet or high fat diet." However, in the Figure legend it is stated: "(E) Gene Ontology analysis of WT and TKO ATM RNA-seq datasets showing the enriched gene ontology terms in TKO ATMs. (F) RNA-seq analysis showing RNA expression in TKO ATMs versus those in WT cells. RNAs up-regulated in TKO ATMs were colored red, whereas RNAs down-regulated were colored blue, gene Tnf was pointed out and colored orange and gene Il10 was pointed out and colored green." Is this data shown in 2E and F now derived from obese or pre-obese animals? If this is derived from just one condition (obese or pre-obese mice), why didn't the authors show all conditions?

In Figure 2E and F, these data were derived from pre-obese animals and a description was added in the figure legend. Those RNA-seq results from obese WT and TKO mice were shown in Figure 2—figure supplement 4A and B.

2) Furthermore, how many animals were used for the RNA-Seq experiment depicted in Figure 2E+F? Is this just 2 samples (one WT and one TKO) pooled from multiple mice? Please either indicate in the figure legends or in the Materials and methods section.

RNA from two mice of each genotype were pooled for RNA-seq experiment and this information was added in figure legend.

3) The authors stated that red/blue marked genes in Figure 2F indicates up- or down-regulated genes. What is their definition for up- or down-regulation? 2-fold change? Or is this based on statistics (which might be problematic with n=1).

In Figure 2F, upregulated and downregulated genes in TKO ATMs were identified as p-value ≤ 0.05, (FPKM +1) fold changes (TKO/WT) ≥ 1.6 for upregulated genes and (FPKM +1) fold changes (TKO/WT) ≤ 0.6 for downregulated genes, which was described in Materials and methods section.

4) Please highlight the top 10 differential expressed genes in the Figure 2E and Figure 5A scatter blot.

Those top 10 upregulated and downregulated genes were highlighted in new Figure 2E and new Figure 5A and were listed in figure legends.